# Noncoding RNAs: Master Regulator of Fibroblast to Myofibroblast Transition in Fibrosis

**DOI:** 10.3390/ijms24021801

**Published:** 2023-01-16

**Authors:** Huamin Zhang, Yutong Zhou, Dada Wen, Jie Wang

**Affiliations:** Department of Immunology, Xiangya School of Medicine, Central South University, Xiangya Road, Changsha 410000, China

**Keywords:** fibroblast, myofibroblast, FMT, ncRNA, signaling pathways, fibrosis

## Abstract

Myofibroblasts escape apoptosis and proliferate abnormally under pathological conditions, especially fibrosis; they synthesize and secrete a large amount of extracellular matrix (ECM), such as α-SMA and collagen, which leads to the distortion of organ parenchyma structure, an imbalance in collagen deposition and degradation, and the replacement of parenchymal cells by fibrous connective tissues. Fibroblast to myofibroblast transition (FMT) is considered to be the main source of myofibroblasts. Therefore, it is crucial to explore the influencing factors regulating the process of FMT for the prevention, treatment, and diagnosis of FMT-related diseases. In recent years, non-coding RNAs, including microRNA, long non-coding RNAs, and circular RNAs, have attracted extensive attention from scientists due to their powerful regulatory functions, and they have been found to play a vital role in regulating FMT. In this review, we summarized ncRNAs which regulate FMT during fibrosis and found that they mainly regulated signaling pathways, including TGF-β/Smad, MAPK/P38/ERK/JNK, PI3K/AKT, and WNT/β-catenin. Furthermore, the expression of downstream transcription factors can be promoted or inhibited, indicating that ncRNAs have the potential to be a new therapeutic target for FMT-related diseases.

## 1. Introduction

Fibroblasts are the major cellular components of connective tissues derived from mesodermal cells. They can secrete extracellular matrix proteins (collagen; non-collagenous glycoproteins such as fibronectin and laminin; proteoglycans) and matrix-degrading enzymes to maintain the homeostasis of the extracellular matrix, proliferating and migrating to wounds to deposit granulation tissue during tissue repair. Generally, cells that secrete the ECM protein, MMPS, TIMPs, and vimentin, but do not express α-SMA, are regarded as fibroblasts (vimentin^+^α-SMA^−^) [1].

Myofibroblasts do not generally exist in normal tissues but are distributed in injured, fibrotic, and cancer tissues. They can exert their functions by ① expressing α-SMA, ② secreting extracellular matrix/collagen, and ③ secreting integrin/cytokine (VEGF/PDGF/TGF-β). The sources of myofibroblasts are complex, involving resident fibroblasts, epithelial cells, endothelial cells, fibrocytes, pericytes, and macrophages [2]. Among these cell types, the differentiation from fibroblast to myofibroblast is considered to be the most important source of myofibroblasts. Myofibroblasts are generally regarded as fibronectin ED-A^+^α-SMA^+^ cells that secrete large amounts of type I collagen and produce matrix cross-linking enzymes. Under normal circumstances, myofibroblasts face three outcomes after completing their function: (1) inactivation and changing into low-activity fibroblasts, (2) becoming scar or aging cells, or (3) apoptosis, through which they are largely eliminated; however, in fibrosis and cancer, myofibroblasts escape apoptosis, continue to multiply, and ultimately cause irreversible damage [1].

FMT is a phenomenon that mainly occurs under pathological conditions and is considered to be the main pathological mechanism of fibrosis. This process goes through two steps: fibroblasts are first differentiated into proto-myofibroblasts containing stress fibers, and the latter turn into mature myofibroblasts containing stress fibers and α-SMA [3]. The causes of FMT can be broadly divided into two categories: physical factors and humoral factors. Physical factors include hypoxia [4], high glucose [5], tension [6], mechanical damage [7], etc. Humoral factors include the stimulation of cytokines, chemokines, growth factors, etc. After receiving these stimuli, the corresponding receptors on the surface of fibroblasts phosphorylate and transmit these signals into the cells through signaling pathways such as TGF-β/Smad, MAPK/P38/ERK/JNK, PI3K/AKT, and JAK/STAT [2], which initiate the expression of related genes and form the myofibroblast phenotype. ncRNAs have been reported to play an important role in the process of FMT. In this review, we summarize the ncRNAs regulating FMT (as shown in Appendix A) and find that ncRNAs mainly affect TGF-β/Smad, MAPK/P38/ERK/JNK, PI3K/AKT, JAK/STAT, WNT/β-catenin, and other signaling pathways, and then promote or inhibit the expression of downstream transcription factors (as shown in Figure 1).

## 2. Overview of Non-Coding RNA (ncRNA)

Among the numerous human genes, non-coding genes account for more than 90%, and a large amount of ncRNA is transcribed. More and more studies have shown that the expression dysregulation of ncRNA is related to a variety of diseases, and ncRNA is expected to become a diagnostic marker and prognostic indicator. There are many kinds of ncRNA, including small nuclear RNA (snRNA), small nucleolar RNA (snoRNA), piwi-interacting RNA (piRNA), small interfering RNA (siRNA), and others. MicroRNA (miRNA), long non-coding RNA (lncRNA), and circular RNA (circRNA) have become the focus of research due to their formidable regulatory capacity.

miRNAs, first identified in the 1990s, originate from genomic introns or exons and are transcribed into the primary transcription of miRNA (pri-miRNAs) under the action of RNA polymerase. In the nucleus, pri-miRNAs are cleaved into pre-miRNAs under the control of Drosha and DGCR8. Pre-miRNAs are subsequently exported to the cytosol by exportin and cleaved into small double-stranded RNAs (dsRNAs) by the Dicer. Finally, dsRNAs bind to Ago2, forming an RNA-induced silencing complex (RISC), and they finally become mature endogenous non-coding single-stranded RNA (18–24 nucleotides in length). miRNAs directly degrade or inhibit target mRNA by complete or incomplete pairing with the 3′ untranslated region (3′UTR) of target mRNA so as to regulate the expression of transcribed genes. Approximately 30% of genes in the human genome are regulated by them [8,9].

lncRNA is a class of endogenous RNA that is more than 200 nucleotides in length but cannot encode proteins due to the lack of a long enough open reading frame [10]. lncRNA contains a 5′-terminal 7-methylguanosine cap (m7G) and a 3′-terminal polyadenylate (poly) tail, so it is easily degraded. lncRNA is transcribed from genomes and is divided into intronic lncRNA, intergenic lncRNA, and antisense lncRNA. It can function in the following ways: ① by remodeling chromatin state, ② by stabilizing proteins or protein complexes, or ③ by acting as competing endogenous RNAs (ceRNA) to regulate gene expression [11]. Due to the development of proteomics and ribosome mapping technology, a small number of lncRNAs have been found to be translated into small molecule peptides through ① CAP-dependent, ② IRES-dependent, ③ M6A-dependent, and ④ SORF-dependent methods, and they play a role in various disease through different mechanisms [12].

circRNA, first identified in plants in 1976 [13], is an endogenous single-stranded circular non-coding RNA with a length from several hundred to several thousand nucleotides derived from pre-mRNA or lncRNA [14,15]. Different from the classical alternative splicing of linear RNA, pre-mRNA undergo reverse splicing during the process of maturation. The 5′ and 3′ ends of pre-mRNA are connected in covalently closed loops to form circRNAs. Circular RNA can resist the degradation of exonuclease precisely because it lacks the free ends of 5′ methyl guanosine and a 3′ poly A tail, which also results in a longer half-life and higher stability than linear RNA [16]. Approximately 30% of circRNAs are conserved among various species, with specific expression in different tissues and pathological stages [17]. circRNAs can be divided into three types according to their components: exonic circRNAs (ecRNA), intronic circRNAs (ciRNA), and exon-intronic circRNAs (eliRNA), among which ecRNA is mostly localized in the cytosol, while ciRNA and eliRNA are localized in the nucleus [18]. circRNA can function in the following ways: ① as an miRNA sponge, ② as an RBP sponge, ③ as a protein scaffold to recruit protein regulation gene transcription and splicing, ④ by being translated into a protein and ⑤ by being secreted into the peripheral blood by exosomes as potential biomarkers [19].

It is worth noting that the mode of action for lncRNA and circRNA is limited by their location. Generally speaking, they are located in the nucleus to act as protein sponges or protein scaffolds directly, while they are located in the cytoplasm by binding to miRNAs, which in turn promotes downstream gene expression. Therefore, when discovering new lncRNA and circRNA, it is important to first focus on their localization in the cell.

## 3. ncRNA Activates or Inhibits Different Signaling Pathways to Regulate FMT

### 3.1. TGF-β/Smad Signaling Pathway

The TGF-β/Smad signaling pathway is recognized as a signal pathway of fibrosis. It has been found that TGF-β1 is upregulated in the fibrosis of the liver, myocardium, lungs, and other organs and induces fibroblasts to differentiate into myofibroblasts through the TGF-β/Smad pathway. Its mechanism of action is as follows: TGF-β binds to TGFβR Ⅱ on fibroblasts, and TGFβR Ⅱ is activated and phosphorylates TGFβR Ⅰ. Phosphorylated TGFβR Ⅰ then phosphorylates downstream Smad2/3, and p-Smad2/3 bind to Smad4 and enter the nucleus as heterotrimers to regulate the transcription of fibrotic genes. This promotes phenotypic change in fibroblasts [20,21].

#### ncRNA in the TGF-β/Smad Pathway

TGF-β/Smad is the most well-known fibrosis signaling pathway by researchers. Multiple studies have shown that ncRNA can target TGF-β, TGFβR Ⅰ/Ⅱ/Ⅲ, and smad2/3/4/7 to regulate TGF-β/Smad and then affect the FMT process.
TGF-β as the downstream target of ncRNA

As the initiator protein of TGF-β/Smad signal transduction, the inhibition of TGF-β can regulate the activation of TGF-β/Smad from the source. circHIPK3 [22], miR-433 [23], miR-425-5p [24], miR-142-3p [25], and miR-29b [26] can regulate the expression of TGF-β to activate or inhibit the TGF-β/Smad signaling pathway. Among them, circHIPK3 and miR-29 have attracted extensive attention from researchers. In 2016, Jeck W R et al. used RNA sequencing analysis and found for the first time that circHIPK3 (has-circ-0000284) was massively expressed in human glans fibroblasts [27]. circHIPK3 has nine direct binding sites and 18 potential binding sites for miRNAs. This means that circHIPK3 can function by binding to different miRNAs [28]. The maternal gene of circHIPK3, HIPK3 (home domain interacting protein kinase 3), is located on human chromosome 11 and is a member of the HIPKs family, which is abundant in human tissues and has a high reverse cleavage rate [29]. The second exon of HIPK3 is more prone to 3′ and 5′ splicing at its canonical splicing site and is driven by the base complementary pairing of ALU repeats in its flanking intron to form circHIPK3. circHIPK3 plays a dual role in cancer [30] and a pro-fibrotic role in fibrotic diseases. In myocardial fibrosis induced by hypoxia, circHIPK3 promotes the expression of TGF-β2, activates TGF-β/Smad, and induces FMT through the competitive binding of miR-152-3p [22]. It also can induce FMT in other ways; for example, in TGF-β-induced pulmonary fibrosis, circHIPK3 ① enhances the expression of FOXK2 by competitively binding to miR-30-3p, promoting the glycolysis level of embryonic lung fibroblasts, induces FMT [31] and ② binds competitively to miR-338-3p, improving the expression of SOX4 and COL1A to induce FMT [32]. There are three members of the miR-29 family: miR-29a, miR-29b (miR-29b-1/miR-29b-2), and miR-29c. Among them, miR-29a is localized in the cytoplasm, while miR-29b/c is localized in the nucleus, which enables miR-29a/b/c to function in different ways [33]. The miR-29 coding gene is adjacent to the CD46/34 coding genes. miR-29a and miR-29b-1 are located in the q32.3 region of human chromosome 7, while miR-29b-2 and miR-29c are located in the q32.2 region of human chromosome 1 [34]. Different members of miR-29 all contain a seed sequence with a length of 7 nt, so their predicted target genes are highly overlapping; however, they each contain unique sequences and play different functions. miR-29 is not expressed or has a low expression in embryonic tissues but is widely expressed in mature tissues. A large number of studies have reported that miR-29 plays a dual role in tumor progression [35]. Its inhibitory effect on fibrosis should not be ignored. For example, Yuan R et al. found that miR-29a in the exosomes of human adipose-derived mesenchymal stem cells (hADSCs) could directly bind to TGF-β2 and inhibit the TGF-β/Smad signaling pathway, thereby inhibiting the formation of pathological scars [36]. lncRNA H19 can target miR-29b to regulate the activity of oral mucosal fibroblasts [37]. Yang J et al. constructed an in vitro silicosis cell model with miR-29c overexpression and inhibition and found that miR-29c could inhibit the SiO2-induced trans-differentiation of lung fibroblasts in vitro [38]. lncRNA TUG1 can competitively bind miR-29c to promote the hypoxia-induced FMT process of cardiac fibroblasts [39]. Thus, circHIPK3 and the miR-29 family are fibrosis therapeutic targets worthy of further study.
TGFβR as the downstream target of ncRNA

TGFβR Ⅰ and TGFβR Ⅱ are important receptor molecules in TGF-β/Smad signal transduction. The inhibition of TGFβR Ⅰ/TGFβR Ⅱ activity can inhibit the activation of the TGF-β/Smad signaling pathway. Previous studies have shown that miR-101a [40], miR-130 [41], miR-let-7i-5p [42], miR-133a [43], miR-125b-5p [44], and lncRNA SNHG20 [45] target TGFβR Ⅰ. miR-30c [46], miR-370 [47], and miR-9-5p [48] target TGFβR Ⅱ. miR-23b-3p [49] and miR-21 [50] target TGFβR Ⅲ. Among these, miR-let-7 has been thoroughly studied. miR-let-7, the first known miRNA in humans, was discovered in Caenorhabditis elegans in 2000. It is about 21 nucleotides in length, and its precursor molecule is a stem-loop folded structure with 70 nucleotides. The miR-let-7 family consists of 13 members, including let-7a, b, c, d, e, f, i, miR-98, and miR-202, which are located on nine different chromosomes [51]. Let7 family members are under-expressed at the embryonic stage but are increased in differentiated cells. Xu C et al. found that miR-let-7i-5p in the exosomes of human umbilical cord mesenchymal stem cells was found to target TGFβR Ⅰ on embryonic lung fibroblasts, the phosphorylation of TGFβR Ⅰ was blocked, the level of phosphorylated Smad3 was reduced, and the TGF-β/Smad signaling pathway was inhibited. The failure of embryonic lung fibroblasts to convert into myofibroblasts alleviates silico-induced silicosis [42].
Smads family as the downstream target of ncRNA

Smad2/3: Smad2/3 plays an indispensable role in the TGF-β/Smad signaling pathway as a downstream molecule that first responds to the TGFβR activation signal. The inhibition of Smad2/3 phosphorylation partially blocks the activation of the TGF-β/Smad pathway. miR-27a-3p [52] targets Smad2, and circAMD1 can competitively bind to miR-27a-3p to promote the proliferation and collagen synthesis of human dermal fibroblasts [53]. miR-370 [47], lncRNA GAS5 [54], and lncRNA CRNDE [55] target Smad3. miR-335-3p [56] targets Smad2/3 simultaneously. lncRNAs GAS5 and CRNDE have attracted the attention of many researchers. lncRNA GAS5, with a gene position of 1q25.1 and a length of 630nt, was first discovered in the stalled mouse fibroblast NIH3T3 [57]. lncRNA GAS5 plays an important role in the occurrence and development of tissue and organ fibrosis, but its role is twofold [58]. For example, in TGF-β-induced myocardial fibrosis and renal fibrosis, the expression of lncRNA GAS5 is decreased in myocardial fibroblasts and up-regulated in renal fibroblasts, playing inhibitory and promoting roles [59,60]. Tang R et al. found that lncRNA GAS5 was down-regulated in both TGF-beta- and bleomycin-treated skin fibroblasts and the overexpression of GAS5 inhibited the TGF-beta-induced activation of fibroblasts. The mechanism of action is as follows: lncRNA GAS5 directly binds to Smad3 and promotes the binding of Smad3 to protein phosphatase 1A (PPM1A) (a Smad3 de-phosphorylase), which dephosphorylates Smad3 and inhibits the activation of downstream pathways [54]. lncRNA GAS5 can also regulate the process of fibrosis through a variety of other pathways, which is a potential therapeutic target for fibrosis. CRNDE (muscle neoplasia differentially expressed), which is located on human chromosome 16 and has a length of 1059nt, plays a dual role in fibrotic disease. The knockdown of CRNDE expression in embryonic lung fibroblasts can alleviate the LPS-induced inflammatory response by inhibiting the phenotype change in embryonic lung fibroblasts [61]. Instead, the overexpression of CRNDE can alleviate synovial fibrosis in rats with osteoarthritis (OA) [62]. Zheng D et al. found that in myocardial fibrosis caused by diabetic cardiomyopathy, the expression of CRNDE was activated by Smad3 and was thus enriched in myocardial fibroblasts; however, increased CRNDE, in turn, inhibited the transcriptional activation of target genes by Smad3, thus inhibiting the differentiation of fibroblasts into myofibroblasts [55]. More studies are needed to elucidate the mechanism of action of CRNDE in FMT.

Smad4: As a central regulator of the TGF-β/Smad signaling pathway, the reduced phosphorylation of Smad4 leads to the inability of Smads to form effective complexes in the nucleus, thereby attenuating or interrupting TGF-β signaling. Both miR-146a [63] and miR-27-3p [64] can directly target Smad4 to regulate Smad4 expression and phosphorylation. miR-146, with a length of 20nt, is the first miRNA considered to regulate the immune system and has attracted the attention of researchers. There are two subtypes of miR-146a and miR-146b, which only have two nucleotide differences at the 3′ end. The former encoding gene is located in the second exon region of human chromosome 5, while the latter is located in the q24–26 region of human chromosome 10 [65]. In myocardial fibrosis caused by myocardial injury, the N-terminal peptide C0–C1f of the myocardial binding protein C can activate the NF-κB pathway through TLR4 to cause inflammation and delay TGF-β-induced fibroblast activation; miR-146 can inhibit the C0-C1f function [66]. These results suggest that miR-146 also plays an important role in fibrosis. Studies have found that the expression of miR-146a is up-regulated in the process of human skin FMT induced by TGF-β1. Bioinformatics predict that Smad4 is the downstream target of miR-146a. Real-time quantitative PCR and Western blot experiments have shown that miR-146a mimics can significantly down-regulate the expression of Smad4 and ultimately affect the generation of α-SMA in myofibroblast differentiation [63].

Smad7: Smad7 negatively regulates the TGF-β/Smad signaling pathway, which can competitively bind to Smad2/3 receptors and prevent Smad2/3 phosphorylation, thereby preventing TGF-β/Smad signaling pathway activation. miR-92a [67], miR-21 [68], miR-877-3p [69], and circHNRNPH1 [70] can affect the TGF-β/Smad signaling pathway by regulating the level of Smad7. In myocardial fibrosis after ischemia, circHNRNPH1 competitively binds to miR-216-5p, induces the expression of Smad7 to accelerate the degradation of TGF-β1, inhibits the migration of myofibroblasts and the expression of α-SMA and col-Ⅰ, and alleviates the progression of myocardial fibrosis.

### 3.2. PI3K/AKT Signaling Pathway

The PI3K/AKT signaling pathway is another important fibrotic pathway that plays an important role in a variety of fibrotic diseases. It has been reported that the PI3K/AKT signaling pathway is a key signaling node in idiopathic pulmonary fibrosis (IPF), and PI3K/AKT inhibitors can inhibit the progression of IPF, which has been verified in clinical trials [71]. Similarly, in oral submucosal fibrosis, PDGF can make oral mucosal fibroblasts differentiate into myofibroblasts through the PI3K/AKT signaling pathway [72]. Its mechanism of action is as follows: extracellular growth factors, such as fibroblast growth factor (FGF), vascular endothelial growth factor (VEGF), and insulin (INS), bind to the corresponding receptors on fibroblasts, activating the G-protein-coupled receptor or receptor complex protein kinase PTK signal, stimulating PI3K to translocate from the cytoplasm to the cell membrane. The activated PI3K catalyzes the phosphorylation of phosphatidylinositol 4.5-diphosphate (PI (4,5) P2) to generate PIP3. AKT and upstream inositol-3-phosphate-dependent protein kinase 1 (PDK1) interact with PIP3 through the PH domain of PI3K. PDK1 activates threonine phosphorylation at 308 within AKT to activate AKT. Phosphorylated AKT regulates the mammalian target of rapamycin (mTOR), hypoxia-inducible factor-1α (HIF-1α), and the reactive oxygen species (ROS) system, which are involved in the transcription of fibrotic genes, finally promoting the phenotypic change in fibroblasts.

#### ncRNA in the PI3K/AKT Pathway

Studies have shown that miR-718 [73], miR-29 [74], miR-126 [75], miR-21 [76], miR-338-3p [77], miR-503 [78], miR-424 [79], lncRNA N341773 [80], lncRNA PFAL [81], lncRNA GAS5 [82], etc., can affect the activation of the PI3K/AKT signaling pathway. Among them, PTEN (phosphate and tension homology deleted on chromosome ten) is a crucial protein regulating the PI3K/AKT signaling pathway. It has lipid phosphatase activity and can dephosphorylate PIP3: the downstream target molecule of PI3K. miR-718 [73], miR-21 [76], and miR-338-3p [77] can directly bind to PTEN, inhibit the expression of PTEN, reduce its inhibitory effect on PI3K/AKT, and promote the progression of fibrosis. miR-21 has been shown to be dysregulated in a variety of diseases. miR-21 is a microRNA with a length of 22 nucleotides that widely exists in eukaryotes. It is the most richly expressed miRNA in the body and also one of the earliest miRNAs discovered. The miR-21 encoding gene is located in the tenth intron region of the q23.2 region of human chromosome 17, close to the vacuole membrane protein 1 (VMP1) gene. The miR-21 sequence has an independent promoter region, and its precursor can initiate transcription autonomously and become mature miR-21. The sequence also contains conserved enhancer elements, including signal transducers and activators of transcription, enhancer-binding proteins, etc., which can improve the transcription level of miR-21. The binding of miR-21 and PTEN has been confirmed in a variety of cell lines. In hypertrophic scars and renal fibrosis, miR-21 induces fibroblasts to differentiate into myofibroblasts through the PTEN/AKT signaling pathway [76,83]. Studies by several teams have proved that lncRNA and circRNA can exert their physiological functions through the competitive binding of miR-21, such as lncRNA GAS5 [84] and circPTPN12 [85].

### 3.3. MAPK Signaling Pathway

In recent years, the role of the MAPK signaling pathway in fibrosis has attracted more and more researchers’ attention. MAPK, a mitogen-activated kinase that phosphorylates both serine/threonine and tyrosine, involves four distinct signaling pathways: ① Ras-Raf-MEK1/2-ERK1/2 ② the JNK cascade ③ the P38 cascade, and ④ the BMK1 (ERK5) cascade, all of which are protein kinase cascades in nature, including three core members: MAPK kinase kinase (MKKK), MAPK kinase (MKK), and MAPK; these three kinases that are activated sequentially regulate a variety of physiological/pathological processes. Among them, Ras-Raf-MEK1/2-ERK1/2 and the JNK cascade are the two most significant MAPK pathways. Cai Z et al. found that BMP4 regulates the proliferation, apoptosis, activation, and differentiation of fibroblasts by inhibiting the ERK/p38 MAPK signaling pathway [86], proving that the MAPK signaling pathway is also involved in the regulation of the fibrosis process. Its mechanism of action is as follows: extracellular growth factors, such as EGF and FGF, bind to and activate tyrosine kinase receptors, such as EGFR and FGFR, on the membrane of fibroblasts, and the latter binds to the adaptor protein GRB2 to recruit guanylate activating factors (SOS) to the cell membrane can convert RAS from inactive RAS-GDP to active RAS-GTP. The activated Raf kinase further binds to MEK1/2 and activates ERK1/2. ERK can enter the nucleus and phosphorylate FOS, MYC, SP1, and other transcription factors to regulate the activation and differentiation of fibroblasts. JNK has three subtypes: JNK1, 2, and 3. Stress signals, such as ultraviolet light, heat shock, cytokines, growth factors, and some G-protein-coupled receptors, can be transmitted to MAPKKK through the Rho subfamily (Rac, Rho), one of the RAS family small molecule G proteins and further activate MEK4/7 and JNK in turn. After activation, JNK can act on a variety of downstream transcription factors (JUN, ELK1, ETS2, etc.) and ultimately promote the expression of fibrosis-related genes.

#### ncRNA in the MAPK Pathway

miR-21 [87], miR-127-3p [88], miR-43 [23], miR-32-5p [89], miR-338-3p [90], miR-155 [91], miR-22 [92], miR-503 [78], lncRNA FENDRR [93], and circEP400 [94] can affect the activation of the MAPK signaling pathway and regulate the differentiation of fibroblasts into myofibroblasts. We found that most of these ncRNAs did not directly target molecules in the MAPK pathway but targeted some proteins regulating the MAPK signaling pathway, such as PDCD4, DUSP1, SPRY1, etc.

PDCD4 (programmed cell death factor 4) can inhibit JNK activity and c-jun phosphorylation and then inhibit the expression and phosphorylation of the eukaryotic translation initiation factor eIF4E. Liao Y.-W. et al. found that miR-21 was up-regulated and PDCD4 was down-regulated in activated oral submucosal fibroblasts. The Luciferase reporter assay proved that miR-21 and PDCD4 have binding sites, so miR-21 directly binds to PDCD4 and partially blocks the inhibitory effect of PDCD4 on JNK activity, thereby promoting the differentiation of oral fibroblasts into myofibroblasts [95].

DUSP1 (dual specificity phosphatases 1), a member of the mammalian protein kinase (MPK) family, is an important negative feedback regulator of the MAPK signal transduction pathway. DUSP1 has phosphatase activity, which mainly dephosphorylates activated MAPK family members, especially tyrosine and threonine phosphate groups in p38 MAPK kinase and JNK kinase, thereby inhibiting the activity of the MAPK pathway. Shen J. et al. treated cardiac fibroblasts with high glucose, and the expression of miR-32-5p was up-regulated, while DUSP1 was down-regulated. The Luciferase reporter assay demonstrated that miR-32-5p could directly target DUSP1, and the MAPK signaling pathway was activated, which promoted the activation and differentiation of cardiac fibroblasts [89].

Spry1 (Sprouty 1) can significantly inhibit the Raf-ERK1/2 and p38 MAPK pathways by inhibiting the phosphorylation of ERK and p38. The study of Li D. et al. showed that, after miR-21 knockdown, Ang-II-induced myofibroblast differentiation was partially inhibited, while the expression of Spry1 was significantly increased, which further inhibited the activation of ERK1/2, indicating that miR-21 could activate the MAPK signaling pathway by targeting Spry1 and finally regulate the fibroblast to myofibroblast transition [96].

In addition, another circRNA affecting the JNK/MAPK pathway is circFgfr2, which also attracted our attention. circFgfr2 is produced by the fibroblast growth factor 2 (FGF2) gene. circFgfr2 competitively binds miR-133 and then promotes the expression of the mitogen-activated protein kinase 20 (Map3k20) gene to activate the JNK/MAPK pathway during skeletal muscle development and myogenesis. The activation of the JNK/MAPK pathway inhibits the activation of the downstream target KLF4. KLF4 can bind to the promoter of circFgfr2 to further improve the expression of circFgfr2. Therefore, a negative feedback regulation loop is formed [97]. Since circFgfr2 can activate the JNK/MAPK pathway and regulate KLF4 expression, it is also likely to affect the FMT process.

### 3.4. Wnt/β-Catenin Signaling Pathway

Wnt/β-catenin signaling is a potentially momentous therapeutic target for a variety of fibrotic diseases. The study of Li X. et al. found that betulinic acid (an anti-tumor and antiviral drug) could inhibit bleomycin-induced pulmonary fibrosis by interfering with the Wnt/β-catenin signaling pathway and inhibiting the differentiation of fibroblasts into myofibroblasts in idiopathic pulmonary fibrosis [98]. The mechanism is as follows: Wnt protein binds to wnt receptors, such as Frizzled (Fz) and low-density lipoprotein receptor-associated proteins on fibroblasts, and can recruit the Dishevelled protein in the cytoplasm to inhibit the activity of the destruction complex (which ubiquitizes and degrades β-catenin) formed by glycogen synthesis kinase 3 (GSK3) and others. This enables β-catenin to accumulate and localize in the nucleus, bind to the TCF/LEF family of transcription factors, and initiate the transcription of downstream fibrotic genes.

#### ncRNA in the Wnt/β-Catenin Pathway

miR-126 [99], miR-33a-3p [100], miR-154 [101], miR-29c [102], miR-142-3p [25], miR-27a-3p [103,104], and lncRNA safe [105] can affect the activation of the WNT/β-catenin signaling pathway and regulate the differentiation of fibroblasts into myofibroblasts. Among them, both miR-33a-3p and miR-154 target DKK to regulate the Wnt/β-catenin signaling pathway. The Dickkopf-associated protein (DKK) is a secreted protein that can form a ternary complex with the transmembrane proteins KREMEN and LRP5/6 to promote the internalization of LRP5/6 and inhibit the binding of LRP5/6 to wnt, thereby inhibiting the Wnt/β-catenin signaling pathway. Henderson, J et al. transfected miR-33a-3p mimics into skin fibroblasts, and then detected the expression of DKK-1 by ELISA and found that miR-33a-3p could activate the Wnt/β-catenin signaling pathway by reducing the expression of DKK-1. Sun and Ly et al. found a direct correlation between miR-154 expression, DKK2 expression, and myofibroblast differentiation. They also demonstrated that miR-154 is directly bound to DKK2. Therefore, miR-154 can activate the Wnt/β-catenin signaling pathway by down-regulating the expression of DKK2 [101].

### 3.5. Other Signaling Pathways

#### 3.5.1. JAK/STAT Signaling Pathway

Papaioannou I et al. found that STAT3 can bind to the COL1A2 gene simultaneously with JunB as an enhancer, recruit RNA polymerase to promote COL1A2 expression and regulate fibroblasts to differentiate into myofibroblasts [106]. Therefore, the JAK/STAT signaling pathway is involved in the FMT process. The mechanism is as follows: JAK is a non-receptor tyrosine kinase. When cytokines (e.g., IL-6, IFN-α/β), hormones, and growth factors (e.g., EGF, PDGF, IGF) bind to the corresponding receptors on fibroblasts (such receptors do not have kinase activity, but their intracellular segment has a binding site for tyrosine kinase JAK), the receptor becomes polymerized. JAK molecules bound to their intracellular segment can be close to each other and need to be phosphorylated to be activated. Activated JAK molecules can phosphorylate STATs, and the phosphorylated STATs undergo dimerization. STAT dimers enter the nucleus and bind to specific regulatory sequences, thus activating the transcription of fibrotic genes. STATs are then dephosphorylated and returned to the cytosol as a monomer waiting to participate in the next signal transduction. miR-210-5p can regulate the JAK/STAT signaling pathway and then regulate the FMT process. Wei, SY et al. found that miR-210-5p was upregulated in human dermal fibroblasts treated with TGF-β. By binding STAT5A and inhibiting its expression, the STAT3 signaling pathway was activated to promote the differentiation of human dermal fibroblasts into myofibroblasts [107].

#### 3.5.2. Notch Signaling

Unlike the above pathways, the Notch signaling pathway is activated only when two cells come into contact with each other because its ligands and receptors are membrane proteins (the ligands of the above pathways are secreted proteins). Yue Z. et al. found that by blocking the Notch signaling pathway in myofibroblasts, the expression of the pro-apoptotic factors Ngfr and Septin4 were up-regulated, the apoptosis of myofibroblasts was accelerated, fibrosis was reversed, and the progression of liver fibrosis was inhibited [108]. Thus, the Notch signaling pathway also plays an indispensable role in fibrotic diseases. Its mechanism of action is as follows: when the Notch ligand and receptor of adjacent cells bind, the Notch protein is spliced three times under the action of integrin metalloproteinase 10 (ADAM10) and γ-secretase, and the active structure of NICD is detached from the Notch protein and released into the cytoplasm and then into the nucleus. It is combined with transcription repressor CSL to inhibit CSL activity and recruit the transcription activator protein family MAML to form an NICD–CSL–MAML ternary complex to promote the transcription of HES, HEY, and other transcription factors, which can promote the expression of downstream fibrotic genes. Several ncRNAs, such as miR-16-5p [109], miR-30d [110], miR-21 [111], and lncRNA HOTAIR [112], can regulate the Notch signaling pathway and affect the fibrosis process. miR-16-5p was down-regulated, and NOTCH2 was up-regulated in the fibrotic tissues of patients with systemic sclerosis. In vitro experiments confirmed that miR-16-5p could directly bind to NOTCH2 and inhibit the expression of collagen and α-SMA, thereby affecting the transformation of fibroblasts. The overexpression of lncRNA HOTAIR in human dermal fibroblasts can down-regulate the expression of miR-34a, activate the Notch signaling pathway, promote the expression of GLI2, and mediate the expression of downstream fibrotic genes. miR-30d and miR-21 have been reported to target jagged1 to regulate the Notch signaling pathway. Jagged1 (JAG1) is a Notch ligand. If the expression of Jagged1 is decreased, the Notch pathway will be blocked or inhibited. The overexpression of the miR-30d attenuated TGF-beta-1-induced proliferation of lung fibroblasts, as well as α-SMA and collagen I expression, is produced by directly binding to the 3’-UTR of JAG1. Restoring miR-30d expression to inhibit the TGF-beta-1-induced activation of JAG1/Notch signaling may be a promising strategy for the treatment of pulmonary fibrosis.

Blocking or inhibiting the activation of signaling pathways and signal conduction is essential to inhibit FMT and alleviate the symptoms of fibrosis. ncRNA can regulate the activation of signaling pathways to a significant degree which, in turn, affects FMT. However, the same signaling pathway is regulated by multiple ncRNAs, which is effective for clinical diagnosis but brings a certain difficulty to drug development. Therefore, finding highly specific ncRNAs is the goal of future research.

## 4. ncRNA Regulates FMT by Directly Affecting the Expression of Nuclear Transcription Factors

Nuclear transcription factors are one of the trans-acting factors, which are important proteins with DNA binding activity. They can initiate the transcription of related genes and regulate their transcription efficiency by binding to the promoters or enhancers of cis-regulatory elements. Cell differentiation and phenotypic switching are mediated by key transcription factors whose functions are finely regulated at the transcriptional, translational, and posttranslational levels. ncRNA regulates the expression of fibrosis-related genes by regulating the expression of transcription factors. Several transcription factors, such as DYRK2, SIRT1, KLF4, NLRC5, CREB, ARID3A, ZEB, SP1, YBX1, and SOX4, have been reported to be regulated by ncRNA. For example, the ZEB family (E-box zinc finger structural protein family) are a necessary transcription factor in embryonic development, including ZEB1 and ZEB2 protein transcription factors. The miR-200 family has been reported to target the ZEB family to regulate FMT. The miR-200 family contains miR-200a, 200b, 200c, miR-141, and miR-429, among which the coding gene of miR-200a/b/miR-429 is located in the p36.33 region of chromosome 1. However, miR-200c and miR-141 were localized in the p13.31 region of chromosome 12. Liao and YW et al. found that the treatment of oral mucosal fibroblasts with arecoline reduced the expression of miR-200b, and the overexpression of miR-200b could target ZEB2 to reduce the expression of the α-SMA gene and thus inhibit the differentiation of mucosal fibroblasts [113]. Lu and My et al., also found that the expression of miR-200c was down-regulated in oral submucosal fibrosis specimens and oral mucosal fibroblasts treated with arecoline. ZEB1 is a direct target of miR-200c, and the decreased expression of miR-200c attenuates its inhibitory effect on ZEB1 and promotes the expression of α-SMA [114].

## 5. Conclusions and Prospective

Endothelial cells, interstitial cells, fibroblasts, and other cells can be activated and differentiated into myofibroblasts with a stronger collagen synthesis ability under physical and chemical stimulations such as inflammation. This promotes the excessive accumulation of ECM in damaged tissues, leading to the formation of fibrosis. In liver fibrosis, hepatic stellate cells can be activated and differentiated into myofibroblast-like cells, which are the main source of liver myofibroblasts. miR-29b was found to inhibit the differentiation of liver stellate cells by inhibiting the activation of TGF-β/Smad [26]. On the contrary, miR-21 can target PTEN to activate the PI3K/AKT pathway and promote the differentiation of liver stellate cells [115]. However, for most fibrotic diseases, the activation of fibroblasts inherent in tissues and their phenotypic changes into myofibroblasts are the main pathological features of their development. In this review, we summarized the ncRNAs that have played an important role in regulating FMT in recent years. miR-21, miR-29, miR-30, miR-let-7, miR-145, miR-146, and miR-200 have received widespread attention from researchers because they can regulate FMT in various ways (as shown in Figure 2 and Figure 3). Multiple lncRNAs and circRNAs have also been reported to regulate FMT by competitively binding to these downstream microRNAs. Compared with biological small-molecule agents, RNA therapy is more accessible to the target. This provides a new method for the diagnosis, treatment, and prognosis of fibrotic disease. However, the following limitations still exist: (1) Some ncRNAs play opposite roles in different diseases, and even their roles in different models of the same disease are controversial. (2) Most of the ncRNAs reported so far are at the tissue and cell level, which still need invasive examination. For the purpose of non-invasive diagnosis, it is of high clinical significance to study the pathogenesis of ncRNAs with an abnormal expression in saliva, urine, and serum. (3) Most of the ncRNA studies are still only in the laboratory stage, and there is still a long way to go before their clinical application. Gallant-Behm et al. found in a clinical trial that the use of miR-29 mimics inhibited collagen expression and the development of fibroplasia in patients’ incisional skin wounds, offering an entirely new treatment for skin fibrosis [116]. However, RNA therapy has problems with immune-related toxicity and other adverse effects, and further optimization of the drug delivery method is needed. 

## Figures and Tables

**Figure 1 ijms-24-01801-f001:**
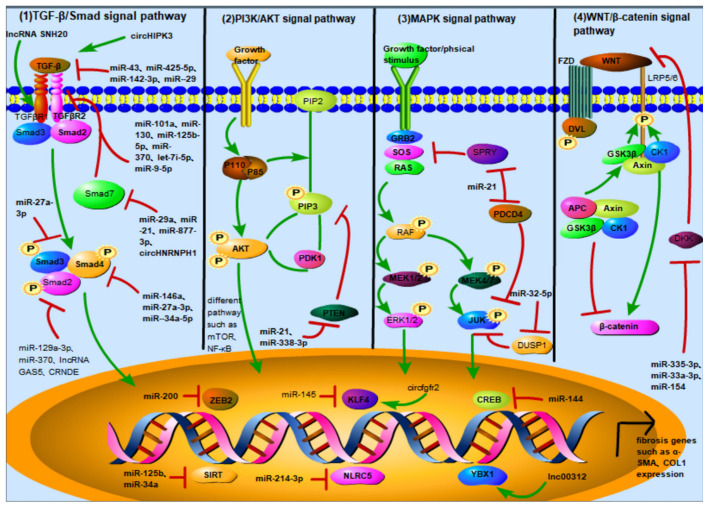
Noncoding RNAs and their targets in different signaling pathways and nuclear transcription factors can affect FMT. (1) TGF-β/Smad signaling pathway: ncRNA can target TGF-β, TGF-βR family, and Smad family to regulate TGF-β/Smad and affect FMT progression. circHIPK3, miR-43, miR-425-5p, miR-142-3p, and miR-29b target TGF-β; miR-101a, miR-130, miR-let-7i-5p, miR-133a, and miR-125b-5p, and lncRNA SNHG20 target TGFβR Ⅰ; miR-30c, and miR-370 and miR-9-5p regulate TGFβR Ⅱ to affect TGF-β/Smad signal transduction; miR-23b-3p and miR-21 target TGFβR Ⅲ; miR-27a-3p targets Smad2; miR-370, lncRNA GAS5, and lncRNA CRNDE target Smad3; miR-146a and miR-27-3p target Smad4; miR-92a, miR-21, and miR-877-3p and circHNRNPH1 target Smad7. (2) PI3K/AKT signaling pathway: PTEN (phosphatase and tension homology deleted on chromosome ten) is an important protein regulating PI3K/AKT signaling pathway. It has lipid phosphatase activity and can dephosphorylate PIP3, the downstream target molecule of PI3K; miR-718, and miR-21 and miR-338-3p can directly target PTEN to regulate PI3K/AKT signaling pathway. (3) MAPK signaling pathway: ncRNA targets some proteins of the MAPK signaling pathway, such as PDCD4, DUSP1, and SPRY1, to regulate the FMT process; miR-32-5p can directly target DUSP1, and miR-21 can simultaneously target PDCD4 and SPRY1. (4) Wnt/β-catenin signaling pathway: Dickkopf-associated protein (DKK) is a secreted protein that can form a ternary complex with the transmembrane proteins KREMEN and LRP5/6 to promote the internalization of LRP5/6 and inhibit the binding of LRP5/6 to wnt, thereby inhibiting the Wnt/β-catenin signaling pathway; miR-33a-3p and miR-154 regulate Wnt/β-catenin signaling pathway by targeting DKK. (5) ncRNA regulates nuclear transcription factors and affects the expression of fibrotic genes: miR-200 targets the ZEB family; miR-125b and miR-34a targeted SIRT; miR-145 targets KLF4; miR-214-3p targets NLRC5; miR-144 targets CREB; and lncRNA 00312 targets YBX1.

**Figure 2 ijms-24-01801-f002:**
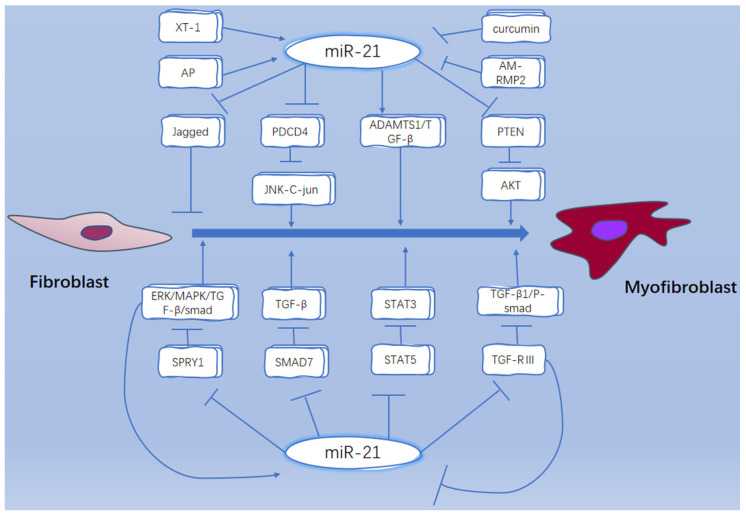
miR-21 regulates FMT through different pathways. miR-21 regulates FMT process by targeting different downstream factors such as jagged1, PDCD4, PTEN, SPRY1, Smad7, STAT5, etc. XT-1 and AP can promote the expression of miR-21, while curcumin and AM-PMP2 can inhibit the expression of miR-21; miR-21 is a potential therapeutic target for fibrosis.

**Figure 3 ijms-24-01801-f003:**
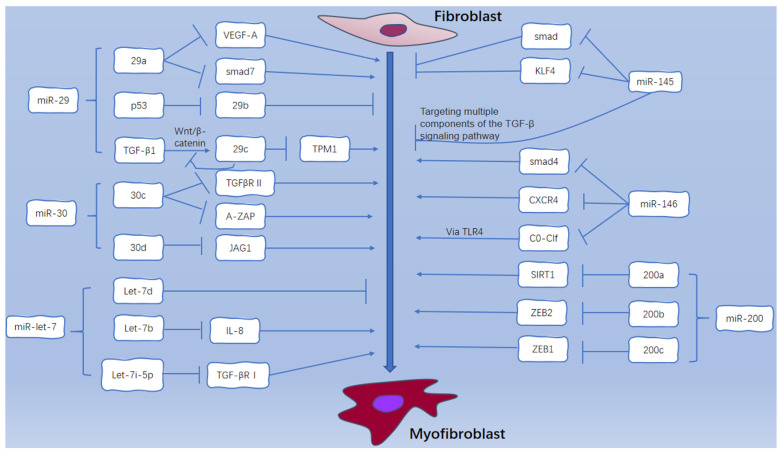
miR-29, miR-30, miR-let-7, miR-145, miR-146, and miR-200 regulate FMT through different pathways: miR-29a targets VEGF-A and Smad7; TGF-β can promote the expression of miR-29c through wnt/β-catenin, and elevated miR-29c in turn inhibits the activation of wnt/β-catenin pathway. miR-30c targets TGF-β R Ⅱ and A-ZAP; miR-30d targets JAG1. let-7b targets IL-8 and let-7i-5p targets TGF-β R Ⅰ. miR-145 targets Smad4 and Smad families. miR-146 targets Smad4, CXCR4 and C0-C1f. miR-200a targets SIRT1, miR-200b targets ZEB2, and miR-200c targets ZEB1.

## Data Availability

No new data were created.

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
