# Peer review of "Noncoding RNAs: Master Regulator of Fibroblast to Myofibroblast Transition in Fibrosis"

_ijms, 2023, doi:10.3390/ijms24021801_

Round 1

Reviewer 1 Report

The major source of myofibroblasts in the liver is non-fibroblasts. There is a fundamental error in understanding. Fibrosis is evaluated histopathologically and is a dynamic pathological condition involving various cells. The author describes the cell biology of myofibroblasts. I suggest changing the title.

Author Response

We sincerely appreciate your careful review and these constructive and insightful suggestions! 

What you mentioned ”The major source of myofibroblasts in the liver is non-fibroblasts“,after reviewing the literature, I found that liver myofibroblasts mainly come from liver stellate cells. Thank you very much for your reminding. And I have deleted the part about liver fibrosis in the text (line 185-188, line 316-320) and separately explained liver fibrosis in the Conclusions and Prospective section.

As for the opinion“Fibrosis is evaluated histopathologically and is a dynamic pathological condition involving various cells”. I totally agree with you, so I restated the origin of myofibroblast cells at the  Conclusions and Prospective section. However, a review of the literature shows that fibroblast to myofibroblast transformation is the main pathological mechanism in most fibrosis diseases(reference:Mechanisms of fibrosis: therapeutic translation for fibrotic disease). My review also adopts this idea. Based on this idea, our review concludes that non-coding RNAs regulate FMT processes through different signaling pathways, hoping to provide some information to researchers in this field, and further promote the diagnosis, treatment and prevention of fibrosis. So we feel that the present title better summarizes the content of this review.

 Based on your kind suggestions, we have rewritten the Conclusions and Prospective section as follows.

Endothelial cells, interstitial cells, fibroblasts and other cells can be activated and differentiated into myofibroblasts with stronger collagen synthesis ability under physical and chemical stimulation such as inflammation. This promotes the excessive accumulation of ECM in damaged tissues, leading to the formation of fibrosis. In liver fibrosis, hepatic stellate cells can be activated and differentiated into myofibroblast-like cells, which are the main source of liver myofibroblasts. Mir-29b was found to inhibit the differentiation of liver stellate cells by inhibiting the activation of TGF-β/smad26. On the contrary, mir-21 can target PTEN to activate the PI3K/AKT pathway and promote the differentiation of liver stellate cells115. However, for most fibrotic diseases, the activation of fibroblasts inherent in tissues and their phenotypic change into myofibroblasts are the main pathological features of their development. In this review, we summarized the ncRNAs that have played an important role in regulating FMT in recent years. Mir-21, mir-29, mir-30, mir-let-7, mir-145, mir-146 and mir-200 have received widespread attention from researchers because they can regulate FMT in various ways (as shown Figure 2 and Figure 3). Multiple lncRNAs and circRNAs have also been reported to regulate FMT by competitively binding to these downstream microRNAs.

Reviewer 2 Report

The manuscript I reviewed was a field review, not a primary data manuscript- so not sure the questions above really apply. The review is clearly written, seems very comprehensive (this is not really my field-- I am not a miRNA/ncRNA expert), but agreed to review because this is what we do as a community.  The strength of this kind of review is that it appears to summarize much of the miRNA/ncRNA literature and targets. Less clear is extent to which targets mentioned are true direct targets. The abundance of information and pathway "targets" makes this review less conceptual. Indeed, I liked the parts of this review where the authors provided more conceptual information about localization informing target choice-- otherwise the review read a bit like 'a list' of information. That said- this will probably be useful to some in the field. 

Author Response

We sincerely appreciate your careful review ! Thank you for your recognition of our work. Based on the idea that fibroblast-to-myofibroblast transformation (FMT) is the main pathological feature of most fibrosis diseases, our review concludes that non-coding RNAs regulate FMT processes through different signaling pathways, hoping to provide some information to researchers in this field, and further promote the diagnosis, treatment and prevention of fibrosis. What you mentioned “Less clear is extent to which targets mentioned are true direct targets”,we rewrote the Conclusions and Prospective section to make it clear.

Compared with biological small-molecule agents, RNA therapy is more accessible to the target. This provides a new method for the diagnosis, treatment and prognosis of fibrotic disease. However, the following limitations still exist: (1) Some ncRNAs play opposite roles in different diseases, and even their roles in different models of the same disease are controversial. (2) Most of the ncRNAs reported so far are at the tissue and cell level, which still need invasive examination. For the purpose of non-invasive diagnosis, it is of high clinical significance to study the pathogenesis of ncRNAs with abnormal expression in saliva, urine and serum. (3) Most of the ncRNA studies are still only in the laboratory stage, and there is still a long way to go before their clinical application. Gallant-Behm et al. found in a clinical trial that the use of mir-29 mimics inhibited collagen expression and the development of fibroplasia in patients' incisional skin wounds, offering an entirely new treatment for skin fibrosis116. However, RNA therapy has problems with immune-related toxicity and other adverse effects, and further optimization of the drug delivery method is needed.

Reviewer 3 Report

Brief summary: This manuscript reviews current information regarding the contribution of noncoding RNA molecules to fibroblast-to-myofibroblast transition (FMT). The manuscript's focus is on the myofibroblasts of fibrotic tissues (rather than cancer). The manuscript defines the categories of noncoding RNAs, then subdivides FMT into well-known regulatory pathways and informs the reader how noncoding RNA molecules participate in these pathways. Figure 1 illustrates most, but not all, of the regulatory pathways. Figure 2 illustrates the complex and complicated way that miR21 regulates FMT, and in a similar fashion figure 3 illustrates 6 other miRNAs' contribution to FMT.

General Concept Comments: This review article provides a complete picture of the up-to-date knowledge of noncoding RNAs in FMT of fibrosis. It builds on knowledge provided in earlier reviews, with more-recent citations added. With exception of two citations (provided below), the list is exhaustive and complete. The manuscript's weakness is in grammatical errors and typographical errors that distract from providing an objective review.

Here are recommendations to increase the merit of this review:

Specific Comments:

Please be consistent with terminology. Regarding FMT in fibrosis, there are two terms most-often used, differentiation, and transition, when discussing FMT. Transformation, on the other hand, is generally reserved for cancers, which is out of the scope of this review. On lines 49, 50, 139, 186, 247, 267, 281, 318, 349, 359, 371, 374, 388, 399, 416, 430, 485, please indicate that fibroblasts differentiate into myofibroblasts, or transition to myofibroblasts.

Fibroblast activation or fibroblast differentiation is separate and distinct from myofibroblast activation or myofibroblast differentiation. Authors please ensure which of these is intended for lines 228, 339, and 368.

Fibroblasts are the resident cells of all connective tissues. Please indicate this in line 27. 

The subject of the manuscript is fibrosis. It will be less distracting to the reader if the authors will remove the cancer background information in lines 204-206, 221-222, 234-240, 310-313. 

Consider using superscripts to designate a positive or negative cell. The information on line 33 and line 41 is unclear as written.

The transition cell in FMT is a proto-myofibroblast. Please make this change in line 49.

There are two kinds of RNA molecules listed, but not defined (line 95, pri-RNA, and line 110, ceRNA). Please define these for reader clarity.

Note also there is no link to figure 3 in the body of the manuscript; additionally, there is some non-English writing on the figure.

While most of the citations are appropriate, the following require changes.

Line 144: 20 and 21 are not appropriate review articles for the provided statement. Instead, authors should use these two effective review articles: Wilson SE, 2021, and Saadat S et al., 2021 (see below). Especially relevant is the latter citation, because it includes earlier but appropriate references to noncoding RNAs that affect TGF-beta/smad signaling in fibrotic myofibroblast differentiation (as does this manuscript), and therefore should be included.

Wilson SE. TGF beta -1, -2 and -3 in the modulation of fibrosis in the cornea and other organs. Exp Eye Res. 2021 Jun;207:108594. doi: 10.1016/j.exer.2021.108594. Epub 2021 Apr 22. PMID: 33894227.

Saadat S, Noureddini M, Mahjoubin-Tehran M, Nazemi S, Shojaie L, Aschner M, Maleki B, Abbasi-Kolli M, Rajabi Moghadam H, Alani B, Mirzaei H. Pivotal Role of TGF-β/Smad Signaling in Cardiac Fibrosis: Non-coding RNAs as Effectual Players. Front Cardiovasc Med. 2021 Jan 25;7:588347. doi: 10.3389/fcvm.2020.588347. PMID: 33569393; PMCID: PMC7868343.

Reference #96 should be 20(21) not 20(1).

Author Response

Thank you very much for your comments and professional advice.These opinions help to improve academic rigor of our article. Base your suggestion and request, we have made corrected modification one by one on the revised manuscript and we have edited articles in English for graphical errors and typical errors. The revision is as follows:

  1. Please be consistent with terminology. Regarding FMT in fibrosis, there are two terms most-often used, differentiation, and transition, when discussing FMT. Transformation, on the other hand, is generally reserved for cancers, which is out of the scope of this review. On lines 49, 50, 139, 186, 247, 267, 281, 318, 349, 359, 371, 374, 388, 399, 416, 430, 485, please indicate that fibroblasts differentiate into myofibroblasts, or transition to myofibroblasts.

We consult a couple of papers to verify this term: differentiation and transformation.There are some examples:

reference1: Myofibroblast differentiation and its functional properties are inhibited by nicotine and e-cigarette via mitochondrial OXPHOS complex III

Reference2:Transforming growth factor (TGF)-β1-induced miR-133a inhibits myofibroblast differentiation and pulmonary fibrosis.

Reference3:Enhanced asthmarelated fibroblast to myofibroblast transition is the result of profibrotic TGFβ/Smad2/3 pathway intensification and antifibrotic TGFβ/Smad1/5/(8)9 pathway impairment.

Reference4:PD-L1 mediates lung fibroblast to myofibroblast transition through Smad3 and βcatenin signaling pathways.

Reference5: Mechanoresponsive regulation of fibroblasttomyofibroblast transition in threedimensional tissue analogues: mechanical strain amplitude dependency of fibrosis.

In the context of fibrosis, both terms have been used by researchers, but differentiation is used only when myofibroblasts are present alone, and the  fibroblasts to myofibroblasts transition appears to be a fixed match, so the following modifications are made:

Revise:

  • lines 49, 50:This process goes through two steps: fibroblasts are first differentiated into proto-myofibroblasts containing stress fibers, and the latter turn into mature myofibroblasts containing stress fibers and α-SMA.
  • Line 139:It is found that TGF-β1 is upregulated in fibrosis of liver, myocardium, lung and other organs and induces fibroblasts to differentiate into myofibroblasts through TGF-β/ SMAD pathway.
  • Line 186:LIANG C et al. found that stimulation of hepatic stellate cells with TGF-β reduced the expression of mir-29b, while overexpression of mir-29b could reduce the expression of smad3 and TGF-β1 and inhibit the differentiation of hepatic stellate cells into myofibroblasts.
  • Line 247:ZHENG D et al. found that in myocardial fibrosis caused by diabetic cardiomyopathy, the expression of CRNDE was activated by Smad3 and thus enriched in myocardial fibroblasts; however, increased CRNDE in turn inhibited the transcriptional activation of target genes by Smad3, thus inhibiting the differentiation of fibroblasts into myofibroblasts.
  • Line 267: Real-time quantitative PCR and Western blot experiments show that miR-146a mimics can significantly down-regulate the expression of SMAD4, and ultimately affect the generation of α-SMA in myofibroblast differentiation.
  • Line 281:Similarly, in oral submucosal fibrosis, the cytokine PDGF can make oral mucosal fibroblasts differentiation into myofibroblasts through PI3K/AKT signaling pathway.
  • Line 318: The same situation also exists in fibroblasts of hypertrophic scars and renal fibrosis, where mir-21 induces fibroblasts to differentiate into myofibroblasts through the PTEN/AKT signaling pathway.
  • Line 349:Mir-21, mir-127-3p , mir-43 , mir-32-5p , mir-338-3p , mir-155 , mir-22 , mir-503 , lncRNA FENDRR , circEP400 can affect the activation of MAPK signaling pathway and regulate the differentiation of fibroblasts into myofibroblasts.
  • Line 359:Luciferase reporter assay proved that mir-21 and PDCD4 had binding sites, so mir-21 directly bound to PDCD4 and partially blocks the inhibitory effect of PDCD4 on JNK activity, thereby promoting the differentiation of oral fibroblasts into myofibroblasts.
  • Line 371:SHEN J et al. treated cardiac fibroblasts with high glucose, and the expression of miR-32-5p was up-regulated, while DUSP1 was down-regulated. Luciferase reporter assay demonstrated that miR-32-5p could directly target DUSP1, and MAPK signaling pathway was activated, which promoted the activation and differentiation of cardiac fibroblasts.
  • Line 374:The study of LI D et al. showed that after miR-21 knockdown, Ang II-induced myofibroblast differentiation was partially inhibited, while the expression of Spry1 was significantly increased, which further inhibited the activation of ERK1/2, indicating that mir-21 could activate MAPK signaling pathway by targeting Spry1 and finally regulate of fibroblast to myofibroblast transition
  • Line 388: The study of LI X et al. found that Betulinic acid (an anti-tumor and antiviral drug) could inhibit bleomycin-induced pulmonary fibrosis by interfering with Wnt/β-catenin signaling pathway and inhibit the differentiation of fibroblasts into myofibroblasts in idiopathic pulmonary fibrosis.
  • Line 399: Mir-126, mir-33a-3p, mir-154, mir-29c, mir-142-3p,mir-27a-3p , lncRNA safe can affect the activation of WNT /β-catenin signaling pathway and regulate the differentiation of fibroblasts into myofibroblasts.
  • Line 416: PAPAIOANNOU I et al. found that STAT3 can bind to COL1A2 gene simultaneously with JunB as an enhancer, recruit RNA polymerase to promote COL1A2 expression, and regulate fibroblasts to differentiate into myofibroblasts.
  • Line 430: By binding STAT5A and inhibiting its expression, STAT3 signaling pathway was activated to promote the differentiation of human dermal fibroblasts into myofibroblasts.
  • Line 485: Liao and YW et al. found that treatment of oral mucosal fibroblasts with arecoline reduced the expression of mir-200b, and overexpression of mir-200b could target ZEB2 to reduce the expression of α-SMA gene and thus inhibit the differentiation of mucosal fibroblasts.

  1. Fibroblast activation or fibroblast differentiation is separate and distinct from myofibroblast activation or myofibroblast differentiation. Authors please ensure which of these is intended for lines 228, 339, and 368.

Revise:

  • line 228: TANG R et al. found that lncGAS5 was down-regulated in both TGF-beta and bleomycin treated skin fibroblasts, and the overexpression of GAS5 inhibited the TGF-beta-induced activation of fibroblasts.
  • line 339: ERK can enter the nucleus and phosphorylate FOS, MYC, SP1 and other transcription factors to regulate the activation and differentiation of fibroblasts.
  • line 368: Luciferase reporter assay demonstrated that miR-32-5p could directly target DUSP1, and MAPK signaling pathway was activated, which promoted the activation and differentiation of cardiac fibroblasts.
  1. Fibroblasts are the resident cells of all connective tissues. Please indicate this in line 27.

Revise:

line 27: Fibroblasts are the major cellular components of connective tissue.

  1. The subject of the manuscript is fibrosis. It will be less distracting to the reader if the authors will remove the cancer background information in lines 204-206, 221-222, 234-240, 310-313.

Revise: lines 204-206, 221-222, 234-240, 310-313 have been deleted from the manuscript, where references 58, 62, 63, 64, 87, and 88 have been deleted at the same time.

  1. Consider using superscripts to designate a positive or negative cell. The information on line 33 and line 41 is unclear as written.

Revise:

  • line 33: Generally, cells secreting ECM protein, MMPS TIMPs and vimentin, but don’t expressing α-SMA, are regarded as fibroblasts(vimentin+α-SMA+).
  • line 41: Among these cell types, the transition from fibroblasts to myofibroblasts is considered to be the most important source of myofibroblasts. Myofibroblasts are generally regarded as fibronectin ED-A+α-SMA+cells that secrete large amounts of type I collagen and produce matrix cross-linking enzymes.
  1. The transition cell in FMT is a proto-myofibroblast. Please make this change in line 49.

Revise:This process goes through two steps: fibroblasts are first differentiated into proto-myofibroblasts containing stress fibers, and the latter turn into mature myofibroblasts containing stress fibers and α-SMA.

  1. There are two kinds of RNA molecules listed, but not defined (line 95, pri-RNA, and line 110, ceRNA). Please define these for reader clarity.

Revise:

  • Line 95: miRNAs, first identified in the 1990s, originate from genomic introns or exons and are transcribed into primary transcription of miRNA (pri-miRNAs)under the action of RNA polymerase.
  • Line 110: acting as competing endogenous RNAs (ceRNA) to regulate gene expression.
  1. Note also there is no link to figure 3 in the body of the manuscript; additionally, there is some non-English writing on the figure.

Revise: Mir-21, mir-29, mir-30, mir-let-7, mir-145, mir-146 and mir-200 have received widespread attention from researchers because they can regulate FMT in various ways (as shown Figure 2 and Figure 3).The correct figure 3 has been re-uploaded.

  1. Line 144: 20 and 21 are not appropriate review articles for the provided statement. Instead, authors should use these two effective review articles: Wilson SE, 2021, and Saadat S et al., 2021 (see below). Especially relevant is the latter citation, because it includes earlier but appropriate references to noncoding RNAs that affect TGF-beta/smad signaling in fibrotic myofibroblast differentiation (as does this manuscript), and therefore should be included.

Revise:These two references have been recited in the manuscript.

  1. Reference #96 should be 20(21) not 20(1).

Revise:Shen, J.;  Xing, W.;  Liu, R.;  Zhang, Y.;  Xie, C.; Gong, F., MiR-32-5p influences high glucose-induced cardiac fibroblast proliferation and phenotypic alteration by inhibiting DUSP1. BMC MOLECULAR BIOLOGY 2019, 20 (21).

Round 2

Reviewer 1 Report

No more comments. Thank you.

Author Response

We sincerely appreciate your careful review.

Reviewer 3 Report

The manuscript is better, but there are still modifications that need to be addressed.

Line 33 should be SMA- not SMA+

Lines 58-62 (60-64 on the marked up version) should be written in present tense.

Line 172 (176 on markup) change from “competitively binding” to “binds competitively” so the verb agrees with “enhances” a few lines earlier.

Line 209 (219 on the markup) there is no citation for Xu et al. It should be citation number 42.

Line 277 (line 296 on the markup) change urging to stimulating

Line 291 (310 on the markup) there is a problem with commas or citation numbers. Please fix.

Line 372 Change L to Li

Line 392-396 (lines 422-425 on the marked up version) sentences need to be rewritten and simplified:

As written: “Sun and LY et al. found that the overexpression or under-expression of mir-154 in cardiac fibroblasts could promote or inhibit the class conversion of cardiac fibroblasts, and down-regulate or upregulate the expression of DKK2. Luciferase reporter assay proved that mir-154 had binding sites with DKK2.”

Change to:

Sun and Ly et al. found a direct correlation between miR-154 expression, DKK2 expression, and myofibroblast differentiation. They also demonstrated miR-154 directly bound to DKK2.

Line 407 change its to their

Line 412 (441 on the marked up version) change “dephosphorylated back to the cytosol” to “dephosphorylated and returned to the cytosol”

Numerous examples of formatting issues with superscripts: lines 52, 153-155, 196-198, 215, 216, 220, 233-34, 258, 261-262, 286-287, 306, 333-334, 360, 383-384, 417- 433-434, (different numbers on the marked up version)

The prefix for micro-RNA molecules is miR. There is some debate on whether the ‘m’ should be capitalized at the beginning of a sentence, so I won’t suggest editing those. However, this manuscript has numerous examples of misspelling miR- prefix as either mir- or Mir-: lines 65, 66, 67, 69, 81, 172-178, 196-198, (many others in between), …504-505, 507-513. (Different line numbers on the marked up version). Most of these might be captured with the “find and replace” editing command.

Also figure 1 numerous examples of mir instead of miR, Lnc and LNC instead of lnc; figure 2 Mir-21 instead of miR-21 in each white oval; figure 3 has Mir instead of miR in each of the white boxes. 

Author Response

Thank you very much for your comments and professional advice.These opinions help to improve academic rigor of our article. Base your suggestion and request, we have made corrected modification one by one on the revised manuscript and our manuscript have completed English editing. The revision is as follows:

  1. Line 33 should be SMA- not SMA+

Revise: Generally, cells that secrete ECM protein, MMPS, TIMPs and vimentin, but do not express α-SMA, are regarded as fibroblasts (vimentin+α-SMA-).

  1. Lines 58-62 (60-64 on the marked up version) should be written in present tense.

Revise: In this review, we summarize the ncRNAs regulating FMT (as shown in Table S1) and find that ncRNAs mainly affect TGF-β/SMAD, MAPK/P38/ERK/JNK, PI3K/AKT, JAK/STAT, WNT/β-catenin and other signaling pathways, and then promote or inhibit the expression of downstream transcription factors (as shown in Figure 1).

  1. Line 172 (176 on markup) change from “competitively binding” to “binds competitively” so the verb agrees with “enhances” a few lines earlier.

Revise: binds competitively miR-338-3p, improving the expression of SOX4 and COL1A to induce FMT.

  1. Line 209 (219 on the markup) there is no citation for Xu et al. It should be citation number 42.

Revise: Xu C et al. found that miR-let-7i-5p in the exosomes of human umbilical cord mesenchymal stem cells was found to target TGFβR Ⅰ on embryonic lung fibroblasts, the phosphorylation of TGFβR Ⅰ was blocked, the level of phosphorylated smad3 was reduced and the TGF-β/SMAD signaling pathway was inhibited. The failure of embryonic lung fibroblasts to convert into myofibroblasts alleviates silico-induced silicosis42.

  1. Line 277 (line 296 on the markup) change urging to stimulating

Revise: Its mechanism of action is as follows: extracellular growth factors, such as fibroblast growth factor (FGF), vascular endothelial growth factor (VEGF), and insulin (INS), bind to the corresponding receptors on fibroblasts, activating the G-protein-coupled receptor or receptor complex protein kinase PTK signal, stimulating PI3K to translocate from cytoplasm to the cell membrane. Activated PI3K catalyzes the phosphorylation of phosphatidylinositol 4.5-diphosphate (PI (4,5) P2) to generate PIP3.

  1. Line 291 (310 on the markup) there is a problem with commas or citation numbers. Please fix.

Revise: miR-71873、miR-2176、miR-338-3p77 can directly bind to PTEN, inhibit the expression of PTEN, reduce its inhibitory effect on PI3K/AKT and promote the progression of fibrosis.

  1. Line 372 Change L to Li

Revise: The study of Li X et al. found that betulinic acid (an anti-tumor and antiviral drug) could inhibit bleomycin-induced pulmonary fibrosis by interfering with the Wnt/β-catenin signaling pathway and inhibiting the differentiation of fibroblasts into myofibroblasts in idiopathic pulmonary fibrosis100.

  1. Line 392-396 (lines 422-425 on the marked up version) sentences need to be rewritten and simplified:

As written: “Sun and LY et al. found that the overexpression or under-expression of mir-154 in cardiac fibroblasts could promote or inhibit the class conversion of cardiac fibroblasts, and down-regulate or upregulate the expression of DKK2. Luciferase reporter assay proved that mir-154 had binding sites with DKK2.”

Change to:

Sun and Ly et al. found a direct correlation between miR-154 expression, DKK2 expression, and myofibroblast differentiation. They also demonstrated miR-154 directly bound to DKK2.

Revise: We have revised it in the manuscript.

  1. Line 407 change its to their

Revise: JAK molecules bound to their intracellular segment can be close to each other and need to be phosphorylated to be activated.

  1. Line 412 (441 on the marked up version) change “dephosphorylated back to the cytosol” to “dephosphorylated and returned to the cytosol”

Revise: We have revised it in the manuscript.

  1. Numerous examples of formatting issues with superscripts: lines 52, 153-155, 196-198, 215, 216, 220, 233-34, 258, 261-262, 286-287, 306, 333-334, 360, 383-384, 417- 433-434, (different numbers on the marked up version)

Revise: We have revised them in the manuscript.

  1. The prefix for micro-RNA molecules is miR. There is some debate on whether the ‘m’ should be capitalized at the beginning of a sentence, so I won’t suggest editing those. However, this manuscript has numerous examples of misspelling miR- prefix as either mir- or Mir-: lines 65, 66, 67, 69, 81, 172-178, 196-198, (many others in between), …504-505, 507-513. (Different line numbers on the marked up version). Most of these might be captured with the “find and replace” editing command.

Also figure 1 numerous examples of mir instead of miR, Lnc and LNC instead of lnc; figure 2 Mir-21 instead of miR-21 in each white oval; figure 3 has Mir instead of miR in each of the white boxes. 

Revise: We have revised them in the manuscript and Uploaded the correct picture again.

Round 3

Reviewer 3 Report

Authors, this version is much better. However please note the following formatting issues to address:  

For example, when RNA molecules have a prefix, it should be lowercase unless at the beginning of a sentence (again, this one is controversial). Examples that should be amended are ecRNA, ciRNA, eliRNA (lines 129-131, 134-6 on the markup version). In the paragraph around line 156 (line 161 on the markup) are numerous circRNAs that are capitalized should be lowercase. LncRNA to lncRNA (line 199; 211 on the markup version)

Smad is found in the manuscript as SMAD, Smad, and smad. Are all these versions correctly used? Please check these to ensure the format is appropriate.

On line 191 (line 202 on the markup version), the citation describes silicon dioxide, not small interfering dioxide. Therefore siO2 should be changed to SiO2.

Line 272 (297 on the markup) change “idiopathic fibrosis” to “idiopathic pulmonary fibrosis”

Line 293 (321 on the markup) replace the 2 unknown, unusual characters (73miR-2176、) with commas.

Line 348 (384 on the markup) change to “member of the mammalian protein kinase (MPK) family”

Author Response

Response

Thank you very much for your comments and professional advice.These opinions help to improve academic rigor of our article. Base your suggestion and request, we have made corrected modification one by one on the revised manuscript. The revision is as follows:

  1. For example, when RNA molecules have a prefix, it should be lowercase unless at the beginning of a sentence (again, this one is controversial). Examples that should be amended are ecRNA, ciRNA, eliRNA (lines 129-131, 134-6 on the markup version). In the paragraph around line 156 (line 161 on the markup) are numerous circRNAs that are capitalized should be lowercase. LncRNA to lncRNA (line 199; 211 on the markup version)

Revise: We've changed all the RNA prefixes to lower case in the manuscript.

lines 129-131, 134-6 on the markup version:

circRNAs can be divided into three types according to their components: exonic circRNAs (ecRNA), intronic circRNAs (ciRNA) and exon-intronic circRNAs (eliRNA), among which, ecRNA is mostly localized in the cytosol, while ciRNA and eliRNA are localized in the nucleus.

line 161 on the markup:

This means that circHIPK3 can function by binding to different miRNAs28. The maternal gene of circHIPK3, HIPK3 (home domain interacting protein kinase 3), is located on human chromosome 11 and is a member of the HIPKs family, which is abundant in human tissues and has a high reverse cleavage rate.

line 199; 211 on the markup version:

Previous studies have shown that miR-101a40, miR-13041, miR-let-7i-5p42, miR-133a43, miR-125b-5p44 and lncRNA SNHG2045 target TGFβR Ⅰ.

  1. Smad is found in the manuscript as SMAD, Smad, and smad. Are all these versions correctly used? Please check these to ensure the format is appropriate.

Revise: We have revised it to Smad in the manuscript.

  1. On line 191 (line 202 on the markup version), the citation describes silicon dioxide, not small interfering dioxide. Therefore siO2 should be changed to SiO2.

Revise: Yang J et al. constructed an in vitro silicosis cell model with miR-29c overexpression and inhibition, and found that miR-29c could inhibit SiO2-induced trans-differentiation of lung fibroblasts in vitro.

  1. Line 272 (297 on the markup) change “idiopathic fibrosis” to “idiopathic pulmonary fibrosis”.

Revise: It has been reported that the PI3K/AKT signaling pathway is a key signaling node in idiopathic pulmonary fibrosis (IPF), and PI3K/AKT inhibitors can inhibit the progression of IPF, which has been verified in clinical trials.

  1. Line 293 (321 on the markup) replace the 2 unknown, unusual characters (73、miR-2176、) with commas.

Revise: miR-71873,miR-2176,miR-338-3p77can directly bind to PTEN, inhibit the expression of PTEN, reduce its inhibitory effect on PI3K/AKT and promote the progression of fibrosis.

  1. Line 348 (384 on the markup) change to “member of the mammalian protein kinase (MPK) family”.

Revise: DUSP1 (dual specificity phosphatases 1), member of the mammalian protein kinase (MPK) family, is an important negative feedback regulator of the MAPK signal transduction pathway.
